# Effectiveness of the Portugal 2020 Programme: A Study from the Citizens' Perspective

**Adriana Z. F. C. Nishimura** [1,*] , **Ana Moreira** [2] , **Manuel Au-Yong-Oliveira** [1,3] **and Maria José Sousa** [4]

1   GOVCOPP (UA), Department of Economics, Management, Industrial Engineering and Tourism (DEGEIT), Campus Universitário de Santiago, University of Aveiro, 3810-193 Aveiro, Portugal; mao@ua.pt
2   Department of Social and Organizational Psychology, University Institute of Psychological, Social and Life Sciences (ISPA-IU), R. Jardim do Tabaco 34, 1149-041 Lisbon, Portugal; amoreira@ispa.pt
3   INESC TEC, Campus da FEUP, University of Porto, R. Dr. Roberto Frias, 4200-465 Porto, Portugal
4   Business Research Unit (BRU-Iscte), University Institute of Lisbon (ISCTE-IUL), Avenida das Forças Armadas, 1649-026 Lisbon, Portugal; maria.jose.sousa@iscte-iul.pt
*   Correspondence: adriana.nishimura@ua.pt

**Abstract:** The European Structural and Investment Funds (ESIF) are the main instrument of the European Union (EU) Cohesion Policy to promote convergence, economic growth and reduce imbalances between EU members. The objectives of the 2014–2020 programming period follow the agenda of the Europe 2020 Strategy to promote smart, sustainable and inclusive growth of EU members. Since before joining the EU, in 1986, until the end of the Portugal 2020 Partnership Agreement (PT2020), Portugal will have received more than EUR 130 billion. Have the subsidies that Portugal has received been well applied? Our study fills a gap in the literature by portraying citizens' perceptions about the effectiveness of EU funds for the development of the country and its regions. The study is quantitative in nature, and a non-probabilistic sample of 1119 participants answered our survey. A high proportion (76%) of the respondents considered that EU funds contributed to the development of the region where they live, although a significant percentage of the respondents (more than half) considered that there may be corruption in Portugal. The Portuguese also mentioned the existence of practices such as favouritism and lobbying regarding the approval of projects. Our findings are supported by the literature, which refers to "lost opportunities" in the inefficient application of ESIF, while recognising that EU funds have played a significant role in Portugal's development over the last three decades.

**Keywords:** Portugal 2020; FEEI; European funds; cohesion policy; development



## 1. Introduction

The European Structural and Investment Funds (ESIF) are the main instrument of the EU Cohesion Policy to support the economic development and convergence of Member States (MS). A total of EUR 454 billion in European funds have been allocated for the 2014–2020 programming period [1]. The ESIF were created to promote the development and the economic and social cohesion of European countries and regions in an integrated, sustainable and harmonious way [2]. To instrumentalise the implementation of the ESIF, the European Commission established a long-term Multiannual Financial Framework (MFF) with the EU members (currently for a period of seven years). The MFF budget is approved by the European Parliament.

Over four MFF (from 1989 to 2013), Portugal received financial support from the ESIF amounting to EUR 107.7 billion [3]. For the PT2020, which runs from 2014 to 2020, the value of the funds approved reached EUR 26.9 billion [4]. These values are substantial and, as a rule, represent between two and three per cent of the Portuguese GDP throughout the funds' programming periods [3].

The programming of PT2020 is aligned with the Europe 2020 strategy for smart, sustainable and inclusive growth of EU members and is organised in four thematic axes: competitiveness and internationalisation; social inclusion and employment; human capital; and sustainability and efficiency in the use of resources. The territorialisation of interventions and the reform of the Public Administration are transversal to these axes. PT2020 has 11 thematic objectives (Table 1).

PT2020 was a break from previous EU frameworks (which placed a strong emphasis on infrastructure) by favouring productive investment and innovation, qualifications and employment in order to promote competitiveness and internationalisation of the economy [3,5]. Another major change was the emphasis on climate change issues with the decarbonisation of the economy and the priority given to clean transport [5].

**Table 1.** Thematic axes and objectives of PT2020.

| | | Thematic Axes |
|---|---|---|
| Transversal axes | Competitiveness and internationalisation | TO 1. Strengthening research, technological development and innovation<br>TO 2. Enhancing access to use and quality of information and communication technologies (ICTs)<br>TO 3. Enhancing the competitiveness of small and medium-sized enterprises-SMEs<br>TO 7. Promoting sustainable transport and removing bottlenecks in network infrastructures<br>TO 11. Enhancing institutional capacity of public authorities and the Public Administration efficiency |
| | Social inclusion and employment | TO 8. Promoting sustainability and employment quality<br>TO 9. Promoting social inclusion, combating poverty and discrimination |
| | Human capital | TO 10. Investing in education and vocational training for skills acquisition and in lifelong learning |
| | Sustainability and efficiency in the use of resources | TO 4. Supporting the transition to a low-carbon economy in all sectors<br>TO 5. Promoting climate change adaptation, risk prevention and management<br>TO 6. Preserving and protecting the environment and promoting efficiency in the use of resources |

Source: Own elaboration. Data from AD&C [6].

The initial allocation of resources of PT2020 foresaw the amount of EUR 25 billion. In terms of thematic areas (Figure 1), the largest percentage was concentrated in the competitiveness and internationalisation axis (41%), followed by sustainability and efficiency in the use of resources (25%). The human capital and social inclusion and employment axes received about 17% each [6].

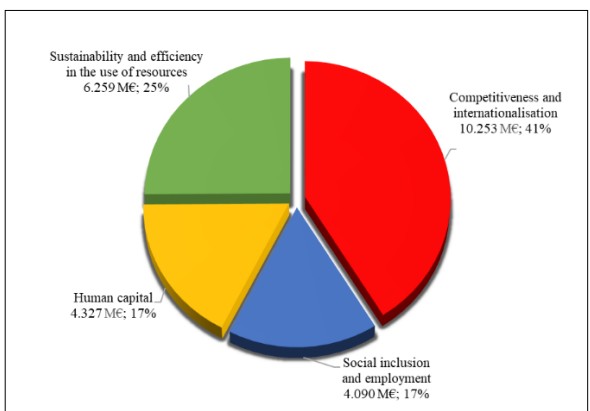

**Figure 1.** Initial allocation of PT2020 resources by thematic areas. Source: Own elaboration. Data from AD&C [6].

As for the PT2020 resources approved and executed by region (Figure 2), by 31 December 2020, the North region absorbed the largest financial volume of funds, amounting to EUR 9 billion, followed by the Central region with EUR 6 billion [4]. The greater

contribution of resources is due to the fact that these regions are eligible for Objective 1 of the regional policy when GDP per capita remains below 75% of the EU average. In addition to the North and Centre, other regions that fall under Objective 1 are Alentejo and the Autonomous Region of the Azores. Algarve is classified as a transition region (GDP per capita between 75% and 90% of the EU average). Lisbon and the Autonomous Region of Madeira have a GDP per capita above 90% of the EU, receiving the classification of more developed regions [5].

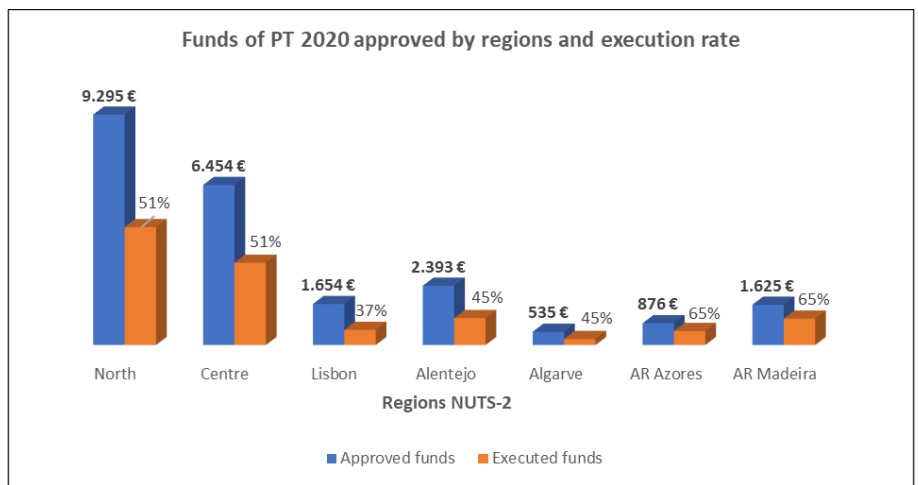

**Figure 2.** PT2020 funds approved and executed by regions. Source: Own elaboration. Data from AD&C [4].

Regarding the execution of funds, the highest rates are in the Azores and Madeira regions, with 65% of the approved budget already executed. At national level, Portugal reached a commitment rate of 104% of approved projects, and an execution rate of 57% [4]. Note that after the approval of the projects, there is a two-year deadline for the execution of the resources, which overlaps with the following financial framework.

The PT2020 quarterly monitoring bulletin, dated 31 December 2020, reports that since the beginning of the programme, in the "competitiveness and internationalisation" axis, over 19,200 companies have been supported in the various incentive systems, and around 4300 R&D and knowledge-transfer projects have been supported. In the axis of "social inclusion and employment" there were more than 1.7 million training actions and around 91,700 beneficiaries of hiring support. In the axis of "human capital", there were more than 117,000 social action scholarship holders in higher education and around 274,000 young people were supported in vocational pathways (in basic and secondary education). In turn, in the area of "sustainability and efficiency in the use of resources", there were more than 145,000 tons of $CO_2$ equivalent related to the annual reduction of greenhouse gas emissions, and more than 9000 households had their energy consumption improved [4].

In terms of European funds approved per capita, an accumulated amount of EUR 2617 per inhabitant was obtained in December 2020, and at the NUTS-2 (Nomenclature of Territorial Units for Statistics, adopted for the EU territories) level, it ranged from EUR 579 (Lisbon region) to EUR 6709 (Azores region). The intensity of support per territorial area ranged from 75,717 €/km$^2$ in the Alentejo region to EUR 1.093 million per km$^2$ in Madeira region, reaching the amount of EUR 291,257 per km$^2$ at a national level [4].

Although the indicators used to assess the application of European funds are continuously monitored by internal and external control entities, and the results of PT2020 point to good performance in its application, the measurement of the impacts of the ESIF on the EU economies is complex. Scientific studies that empirically assess the effectiveness of European funds for the development and convergence of EU members, usually represented by econometric analyses, are not consensual [5,7–12].

Malakhova et al. [13] state that the economic imbalances between the most developed countries (Nordic and North-Western European countries) and the least developed (Southern and Central and Eastern European countries) put the European integration project at risk and hinder the process of convergence between countries. Additionally, they [13] reveal a tendency of the public debt of almost all the southern European countries (including Portugal), which are hardest hit by the global economic crises, to grow.

Thus, it is important to analyse the impacts of the ESIF on the Portuguese economy; whether these impacts are sustainable over time; the stage of development reached by the country (historically a more vulnerable economy in the European scenario); and how these impacts are perceived by Portuguese citizens—the main focus of our study.

The main objective of this study is to know the perceptions of Portuguese citizens about (a) the existing information about the PT2020 Programme; (b) the credibility of the programmes financed by European funds for the development of Portugal and its regions; and (c) the degree of confidence in the adequacy of the fund allocation to Portugal and in the meritocracy of the approved projects.

As secondary objectives, the study aims to find out how citizens evaluate the objectives of the Europe 2020 Strategy and what factors negatively impact the future of Portugal and the EU.

Starting from some variables adopted in instruments for evaluating public opinion about the application of EU funds, we propose a brief conceptual model (Chart 1) in which the perception of the positive effects of ESIF for the development of an EU member is expected to be closely related to the fact that people recognize the Partnership Agreement (PA) brand, identifying it from the outset as an EU program. In turn, the perception of the general positive effects of ESIF should also be positively associated to the belief that ESIF contribute to the economic growth of regions. Regarding meritocracy in the application of ESIF, the perception that there may be corruption in the allocation of funds might be related to a lower confidence in the adequacy and transparency in the approval of ESIF-funded projects.

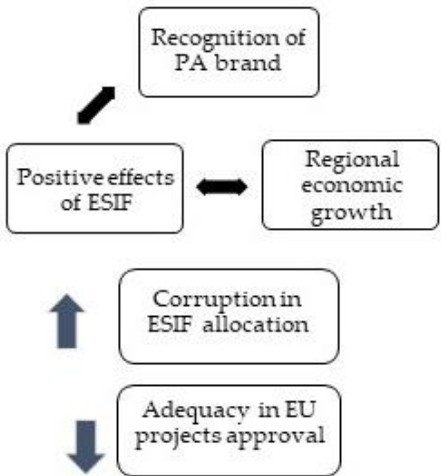

**Chart 1.** Model of ESIF evaluation for public opinion. Source: Own elaboration.

Due to test the assumptions proposed in this model, we formulated the following research hypotheses:

**Hypothesis 1 (H1).** *The credibility of the contribution of programmes financed by European funds to Portugal's development is not independent of the recognition of the Portugal 2020 partnership agreement; participants who are familiar with this agreement are expected to have higher levels of credibility.*

**Hypothesis 2 (H2).** *The credibility of the contribution of the programmes financed by European funds for the development of Portugal is not independent of the fact that, considering that the resources of Portugal 2020 are important for the economic growth of their region and their district,*

*the participants who believe more in these programmes are also those who consider that these resources are important for the economic growth of their region and their district.*

**Hypothesis 3 (H3).** *The degree of trust in the approval of projects financed by European funds, in relation to the adequacy and transparency of the participants is not independent of the fact that in believing that there may be corruption in the allocation of European funds to Portugal, it is expected that the degree of trust in the approval of projects financed by European funds, in relation to the adequacy and transparency of the participants who believe in the existence of corruption, is lower than that of those who do not believe in the existence of corruption.*

For this research, data were collected through an online survey and disseminated nationally in the period from June to December 2020. A total of 1119 complete responses were obtained. For statistical treatment of the data, *SPSS Statistics 25* software was used. For hypothesis testing, inferential statistics (chi-square test) was applied.

The results of the study, which will be reported and discussed ahead, reveal a positive evaluation of the European funds for the development of Portugal and its regions, although a reasonable percentage of respondents consider that there may be corruption or practices such as "lobbying" in the allocation of funds. Although there is a large allocation of PT2020 funds in the competitiveness and internationalization axis, the axis that presented the highest relevance for the citizens is human capital.

Following this introduction section, we will summarise some studies on the impacts of ESIF on the development of EU members and Portugal. The analysis of this literature is important to understand how the thematic objectives of the Europe 2020 Agenda were designed to meet the dynamics of the EU Cohesion Policy over time and will provide holistic and transversal inputs to better interpret our findings.

The third section presents the methods employed in our study, followed by the results obtained and the discussion of these results. In the concluding section, the contributions of our study are to be found, as well as its limitations and suggestions for future work.

## 2. Impacts of ESIF on the Development of EU Members and Portugal

Baleiras [14] states that the Cohesion Policy brings an extremely significant added value to the EU and each MS; at the political level, it is the best expression of the constitutional value of solidarity. The Cohesion Policy aims at economic development, due to the full exploitation of the wealth-generating capacity of each territory. However, the stimulus does not simply involve distributing money to agents and beneficiaries; subsidies are only justified if the results increase social welfare. The gain for society should exceed the private gain of the agent that receives the support, even if it is in the medium and long term [14].

The studies that empirically analyse the efficiency of European funds for the development and economic growth of EU members in order to converge with each other are not consensual [5,7–12]. As a rule, these studies use in their methodologies econometric analyses on EU members growth and convergence between European regions over a given period (which often coincides with the programming periods of the funds), and adopt indicators, such as GDP Per Capita, Employment, Productivity, Investment Rate and Gross Value Added (GVA).

According to Dall'erba and Le Gallo [15], the lack of a common result among studies is partly due to differences in samples (number of countries or regions), the period of analysis of the studies and the econometric techniques used (panel, time series, cross-section, lags, country effects and others).

Still, some studies highlight positive effects on the economic growth of EU members and regions [5,8,10–12,16–29], although there are some conditionalities, as we will see ahead.

Conversely, the effects found in other studies [15,30–36] are null or very insignificant for economic growth and convergence of EU regions.

In Table 2, we present a summary with the territorial coverage, programming period and main results of some of these empirical studies about ESIF effects on EU members.

**Table 2.** Empirical studies on the impacts of ESIF.

| Authors | Territorial Coverage | Period of Programming | Main Results |
|---|---|---|---|
| Oliveira and Leitão [5] | Norte 2020 Programme (Portugal) The Norte region (NUTS-2) is Objective 1 | 2014–2020 | The competitiveness and internationalisation axis had the smallest impact on GVA per euro invested. In terms of job creation, there were positive effects with the incentives allocated to human capital (greater multiplier effect) and social inclusion, in the medium term. |
| Maynou et al. [8] | 174 NUTS-2 regions in the 17 countries of the euro area | 1990–2010 | Positive effects of the funds on economic growth in beneficiary regions and on convergence between countries of the euro zone, with a reduction in disparities. The funds have contributed positively to increase the GDP per capita of the countries. |
| Gagliardi and Percoco [10] | 1233 EU NUTS-3 regions, although the eligibility criterion for Objective 1 is NUTS-2 | 2000–2006 | Cohesion funds have made a positive contribution to generating economic growth in lagging areas. The effectiveness of the funds is greater in rural areas close to urban centres. |
| Tavares et al. [11] | 278 municipalities in mainland Portugal | 2003–2010 | A higher number of fund items accessed by municipalities is associated with higher inflows of firms, higher net inflows and lower outflows. The amount of funds has a weaker association with entrepreneurial dynamics. The amount is associated with an increase in unemployment (especially of short-term), and the number of items is associated with a decrease in unemployment, but both are statistically insignificant. |
| Arbolino and Boffardi [12] | NUTS-2 and NUTS-3 regions of Italy | 2007–2015 | The combination of cohesion investments, with the Institutional Quality Index-IQI, are positively related to overall regional growth. Efficiency indexes are important key factors mainly in the Southern region of Italy, characterised by structural weaknesses. |
| Dall'erba and Le Gallo [15] | 145 NUTS-2 regions of the EU-15 | 1989–1993 1994–1999 | There was significant convergence between the regions studied, but no evidence was found that the funds impacted on this convergence. The investments directed to the peripheral regions have no impact on neighbouring regions. |
| Crescenzi and Giua [16] | EU-wide Regions NUTS-3 of Germany, Italy, Spain and the United Kingdom | 2000–2010 2010–2014 | EU Cohesion Policy has a positive EU-wide impact on regional growth and employment, but it works differently among MS. Very heterogeneous, country-specific economic impacts were found. While Germany achieved the highest growth, the U.K. had the best performance in terms of employment rate. Italy benefited from a short-time positive impact on employment rate, but it vanished after the crisis. In contrast with the rest of the EU, funds fostered a higher growth in Spain after the crisis but did not impact the employment rate. |
| Butkos et al. [17] | 1251 NUTS-3 regions of the EU-25 244 NUTS-3 control group (non-beneficiary) regions | 2000–2006 | The impact of regional support on convergence is positive, with the marginal effect decreasing as payment intensity increases. The returns are larger for the post-intervention period, i.e., the convergence results of EU support occur in the long run. |
| Becker et al. [18] | EU NUTS-2 regions (except Bulgaria, Croatia and Romania) Objective 1 | 1989–1993 1994–1999 2000–2006 2007–2013 | Positive immediate effects on income per capita, but not lasting beyond the programming period of the funds. Negative effects when regions lose Objective 1 eligibility. Effects are weakest during economic crises, especially in hardest hit countries. |
| Crescenzi and Giua [19] | 139 NUTS-1 and NUTS-2 regions, from 12 EU-15 countries (except Denmark, Ireland and Luxembourg) | 1995–2013 | Positive impacts of fund transfers on regional growth rates. Rural development investments are not explicitly linked to regional growth. The impact of the EU Cohesion Policy is strongest in the most supplied regions, and when it is accompanied by Common Agricultural Policy (CAP) funds. |
| Becker et al. [20] | 2078 NUTS-3 regions in the EU Objective 1 | 1994–1999 2000–2006 | Positive effects, but they depend on a high degree on the absorptive capacity of the funds (measured by the quality of regional institutions and the stock of human capital). Only 41% of regions (those with good enough human capital and institutions) are able to transform fund transfers into faster growth. |
| Fiaschi et al. [21] | 173 NUTS-2 regions in the EU-12 | 1980–2002 | The structural funds have had an overall positive effect on productivity growth, but the Objective 1 funds are the main contributor to these results. The effects were most evident in the 1989–1993 and 1994–1999 programming periods. |
| Puigcerver-Peñalver [23] | 41 NUTS-2 regions in the EU-8 Objective 1 | 1989–1993 1994–1999 | The funds had positive effects on the growth of Objective 1 regions in the first period (1989–1993). In the second period (1994–1999), the effect of the funds was zero or even negative. |
| Rodriguez-Pose and Fratesi [26] | 152 NUTS-2 regions in EU-8 Objective 1 | 1989–1993 1994–1999 | Funds spent on infrastructure and institutional support did not significantly impact convergence objectives. Investments in education and human capital showed positive, medium-term effects. Agricultural subsidies were conducive to positive effects on growth but short termed. The effects of the funds were most significant between 1994 and 1999. |
| Mohl and Hagen [28] | 126 EU-15 regions NUTS-1 and NUTS-2 | 1995–1999 2000–2006 | Only the funds allocated under Objective 1 promoted economic growth in the regions. |

**Table 2.** *Cont.*

| Authors | Territorial Coverage | Period of Programming | Main Results |
|---|---|---|---|
| Simões et al. [29] | 30 NUTS-3 regions of Portugal | 1995–2007 | There is a negative relationship of ESIF with productivity at regional level. It was found a positive relationship between human capital and productivity. |
| Ederveen et al. [30] | EU-13 | 1965–1990 | The funds have been shown to be ineffective for the average country. Structural funds are only efficient if they are allocated in countries with good governance capacity, based on transparency, control of corruption and quality of Institutions. |
| Esposti and Bussoletti [32] | 206 NUTS-2 regions of the EU-15 | 1989–2000 | The impact of the funds on the economic growth of Objective 1 regions is rather limited and may become insignificant or even negative if regions are grouped by country (this was observed in Germany, Spain and Greece). |

Source: Own elaboration.

In Portugal, there are few articles that assess the impacts of the European funds applied specifically in the 2014 to 2020 programming (PT2020) for the development of the country and regions, possibly due to the fact that the agreement is still in execution. Some articles refer to EU financial frameworks prior to PT2020, such as the National Strategic Reference Framework (NSRF), ESIFIII or even before.

Certain studies carry out a longitudinal analysis on the effects of the ESIF on the socio-economic development of Portugal over three decades of integration into the EU [3,37,38]. Mateus [37] considers that the ESIF were the main benefit of Portuguese integration into the EU, due to all the influence they had, both in public and private investments, in addition to the contribution of the ESF to the vocational training of the Portuguese. Despite this, the author criticizes the way that the funds were managed, with an excessive fragmentation of projects, the orientation towards infrastructures to the detriment of actions for the qualification of human resources and management capacity, the preference for individual projects, and wasting possible synergies.

In turn, Pires [3] attributes the modernisation of Portugal to the 30 years of support from European funds, which resulted in "remarkable economic and social progress" (p. 37), which provided a better quality of life throughout the country. On the other hand, the author questions whether, in the absence of rigorous analyses prior to the investments and benefiting from high non-refundable co-financing, dispensable investments were made, and others were oversized.

Marques [38] differs somewhat from the others, for whom the strengthening of competitiveness and the convergence of the Portuguese economy to the level of the most advanced EU countries were well below expectations, despite the interventions having made a significant contribution in terms of social inclusion. He attributes the failure to an institutional deficit, generated by the lack of experience in structural policy and the absence of consistent policy options to programme and apply the funds efficiently.

Oliveira and Leitão [5], Tavares et al. [11], and Simões [29] used econometric approaches in their studies, and are referred to in Table 2. Overall, although positive effects of ESIF were found, it follows a non-consensual trend. The regional programme NORTE2020 had an impact on the national GVA in the order of EUR 4 billion and led to the creation of 112,000 jobs. The axes "human capital" and "social inclusion and employment" were the ones that presented the most significant multiplier effects in the medium term [5]. Tavares et al. [11] studied the impacts of ESIFIII and NSRF funding on business dynamics and employability at the local level (municipalities of mainland Portugal). The findings indicated that the larger number of items funded have a greater impact than the amount of funds. Hence, between accessing a larger volume of funds or the wider coordination of funds, one should choose the latter if the objective is to boost local economies [11]. Simões et al. [29] studied the relationship between inequality and economic growth and found a negative association of ESIF with productivity at the regional level, which the authors attribute to the fact that EU funds have been a source of the "Dutch disease" (concept created by *The Economist* in 1977, according to which the exploitation of natural resources

leads to a decline in the manufacturing sector, due to exchange-rate appreciation making exports of manufactured goods less competitive) for Portugal, which resulted in a lack of external competitiveness of the Portuguese economy due to excessive specialisation in non-tradable goods. In contrast, the findings of the study confirm the positive relationship between human capital and productivity [29].

Medeiros [39] analysed projects approved under PT2020 funding for the "sustainability and efficiency in the use of resources" axis, whose initially planned share in the PA has been reduced from 25% (in 2014) to around 14% (in 2019). The results point out that PT2020 funding has contributed to the promotion of sustainable territorial development, namely in the fields of environmental protection and social environmental awareness. However, they fail in not promoting a strategic exploration of renewable energies and of the circular economy, aligned with territorial potentialities [39].

Caldas et al. [7] studied the influence of sustainability on the impact of the ESIF for Portuguese municipalities, along ESIFIII and NSRF frameworks. The main results are that in both programmes, the difference in the impact of investment was not significant, regardless of whether a municipality had higher or lower levels of sustainability. Furthermore, both the efficiency and productivity of investments were higher in ESIFIII than in NSRF for sustainable and non-sustainable municipalities.

In another study, Medeiros [40] examined, over two decades, the territorial impacts of the EU Cohesion Policy in Portugal at the national level. His findings indicate that there have been positive impacts, namely by supporting social cohesion aspects, such as education and training, as well as the construction of cultural, health and sanitary infrastructures. On the other hand, the Cohesion Policy has failed to establish a long-term, integrated territorial development strategy. For example, the lack of spatial planning in the implementation of key infrastructures ran counter to the objective of developing a more polycentric territory at the national level. The concentration of most investments in the two largest agglomerations (Lisbon and Porto) intensified the bi-centric territorial pattern, generating negative effects in the acceleration of a chaotic and sparse urbanisation [40].

Cordeiro et al. [41] researched the application of NSRF funding for the requalification of pre-school and elementary schools of 19 municipalities that integrate the CIMRC (Coimbra Region Intermunicipal Community), which present a very heterogeneous profile in terms of population density. No direct relationship was observed between the approved investments in educational equipment and the characteristics of the territory. Even so, according to the authors, the ESIF provided a true transformation in the Portuguese educational park, which was totally "out of tune with the reality of the 21st century". The next ESIF programming (PT2020) should concentrate investments of immaterial nature in education [41].

Walheer [42] has measured EU members' achievement of the Europe 2020 agenda targets of inclusive, smart and sustainable growth (Table 3). Data were collected from the Eurostat database and refer to the EU-28 over 12 years: 2004 to 2015.

Portugal appears as one of the countries that showed significant growth in achieving targets. In the smart growth pillar, it increased from a score of 0.17 (2004) to 0.51 (2015), but still under the EU-28 average, in 2015 (0.6). In the sustainable growth pillar, it increased from a score of 0.57 (2004) to 0.84 (2015), also under the EU-28 average in 2015 (0.93). It was in the inclusive growth pillar that Portugal performed best, going from 0.42 (2004) to 0.85 (2015), almost reaching the EU-28 average, in 2015, of 0.86 [42].

Regarding how society perceives the allocation and application of European funds in Portugal, there is a lack of empirical studies from the point of view of citizens.

For the elaboration of the Common Communication Strategy of PT2020, in 2013 the Portuguese Agency for Development and Cohesion (AD&C) applied an opinion survey to evaluate the results of the communication strategy of the previous community framework, the NSRF [43]. A total of 88% of the respondents considered that EU funds have been important for Portugal, 74% recognised that the funds have contributed to the development of the country and 66% considered that funds positively impacted the development of the

regions. On the other hand, 43% of the surveyed population considered that there was a good application of the funds, 29% considered the notoriety of the NSRF brand to be good, and only 13% considered that there was sufficient information on the European funds in force and their form of application [43].

**Table 3.** Targets of the pillars of the Europe 2020 Agenda [42].

| Pilar | Topic | Goal |
|---|---|---|
| Smart growth | Higher education R&D | Ensure at least 40% of young adults (aged 30–34) have a university degree or equivalent. Increase investment in R&D to 3% of the GDP. |
| Sustainable growth | $CO_2$ emissions Renewable Energies Energy efficiency | Cut greenhouse gas emissions by 20% Ensure that 20% of energy needs are supplied by renewable sources. Increase energy efficiency by 20%. |
| Inclusive growth | Employment School dropouts Poverty | Raise the employment rate of the population aged 20–64 to 75%. Reduce early school leaving to under 10%. Reduce the number of people living below the poverty line by 25%. |

Source: Adapted from Walheer [42].

In order to improve these indicators, some of the strategic objectives of PT2020´s Communication policy were to increase the positive perception of the funds' application and their impacts on the development of cities and regions; to guarantee the existence of sufficient information on the funds application; and to increase the notoriety and recognition of the PT2020 brand in relation to the NSRF brand [43].

According to the PT2000 Global Evaluation Plan [44] the public opinion survey to assess perceptions about the effects of PT2020 is planned to be applied in the second semester of 2021.

Recently the European Commission [45] applied the Eurobarometer Winter 2020/2021 to poll the public opinion of the citizens of each MS regarding the EU, the national economic and political context and the impacts of the coronavirus pandemic. In Portugal, 1100 citizens were surveyed. When asked about the most important problems affecting the country, 52% of Portuguese citizens chose the economic situation in first place, followed by unemployment (40%) and health (32%). Only 10% of Portuguese people rated the national economy as positive. The panorama in Portugal, as in most Southern European countries, contrasts sharply with that of countries such as Sweden and Luxembourg, where more than 80% of respondents consider their economic situation to be good or very good.

In this survey [45] Portugal stands out as the MS with the highest rates of confidence in European institutions (European Parliament, European Council, European Commission and European Central Bank), with an average of 81%, much higher than the average for EU-27 (47%). Furthermore, 84% of the Portuguese rejected the idea that the country could better face the future outside the EU, with an increase of 12 percentage points, compared to the previous survey conducted in the summer of 2020. On the other hand, the trust of the Portuguese in national institutions is much lower. In central government, it has fallen from 52% to 38%; in regional and local authorities, the trust rate is 52%; and in political parties, it is only 15%.

As for the pandemic, 100% of the Portuguese inquired consider that it has brought serious economic consequences for the country, and Portugal was the EU-27 member whose population least believes in national economic recovery (82% of respondents believe that it will only get some recovery in 2022).

## 3. Methods

### 3.1. Procedure

A total of 1119 participants collaborated in this study, all of whom met the necessary conditions to be considered in the subsequent analyses of it (being over 18 years old and

a resident in Portugal). The sampling process was a non-probabilistic, convenience and intentional *snowball* type of sample [46,47].

The survey was placed online in the Forms UA platform. The questionnaire contained the necessary information on the purpose of the study, ensuring the confidentiality of the answers. It was composed of six sociodemographic questions and seven other questions about the impact of European funds for Portugal. Data were collected between the months of June and December 2020.

### 3.2. Participants

The participants in this study are aged 18 years or more, 129 (11.5%); between 18 and 22 years, 127 (11.3%); between 23 and 29 years, 106 (9.5%); between 30 and 36 years, 211 (18.9%); between 37 and 43 years, 230 (20.6%); between 44 and 50 years, 173 (15.5%); between 51 and 57 years, 101 (9%); between 58 and 64 years, 31 (2.8%); between 65 and 70 years and 11 (1%) more than 70 years. Regarding gender, 529 (47.3%) participants belong to the male gender, 585 (52.3%) to the female gender and 5 (.4%) to another gender.

With regard to academic qualifications, 145 (13%) of the participants have up to the 12th grade, 88 (7.9%) have a university degree, 293 (26.2%) a bachelor's degree, 316 (28.2%) a master's or postgraduate degree, 219 (19.6%) a doctoral degree, 31 (2.8%) a post-doctoral degree, 23 (2.1%) an aggregation degree and 4 (0.4%) another level of education.

In relation to nationality, 1080 (96.5%) are Portuguese and 39 (3.5%) are of another nationality (mainly Brazilian). The participants reside in all the districts of mainland Portugal and the Autonomous Regions, with the highest percentage residing in the districts of Aveiro, Lisbon and Porto, according to Figure 3.

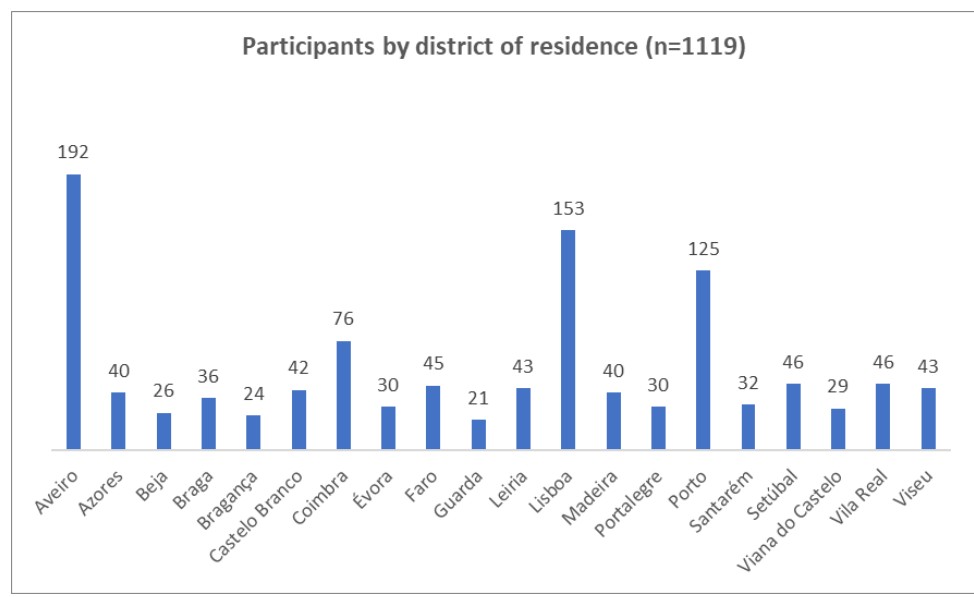

**Figure 3.** Participants by district of residence. Source: Own elaboration.

In terms of profession, 209 (18.7%) are students, 289 (25.9%) are teachers, 58 (5.2%) are researchers, 329 (29.4%) are civil servants, 140 (12.5%) are employees, 38 (3.4%) are self-employed, 11 (1%) are unemployed, 20 (1.8%) are retired and 25 (2.2%) have another profession.

### 3.3. Instrument

The questionnaire consists of 6 socio-demographic questions: age, gender, nationality, district of residence, academic qualifications and profession.

Then there were 7 questions about the importance and use of European funds in Portugal, such as the following: do you know Portugal 2020?; how much do you believe that

the programmes financed by European funds contribute to the development of Portugal?; do you consider that the resources of Portugal 2020 are important for the economic growth of your region and your district?; in your opinion, what are the most important objectives defined for the Portugal 2020 Program?; how much do you trust the approval of projects financed by European funds, regarding adequacy and transparency?; do you believe there may be corruption in the allocation of European funds to Portugal?; and what factors may negatively interfere in the future of Portugal and the European Union? The answers to some questions were randomised in order to avoid bias, characteristic of these types of questions, where respondents tend to choose the first options on the dropdown list.

The questions "Do you know Portugal 2020?" and "Do you consider that the resources of Portugal 2020 are important for the economic growth of your region and your district?" had as response options "yes", "no" and "not sure". The question "Do you believe that there may be corruption in the allocation of European funds to Portugal?" had the response options "yes", "no", "not sure" and "other".

The question "How much do you believe that the programmes financed by European funds contribute to Portugal's development? "is a five-point Likert-type ordinal question ranging from 1 "very little" to 5 "a lot". The question "How confident are you in the approval of projects financed by European funds, with regard to their adequacy and transparency?" is also an ordinal Likert-type question ranging from 1 "very low" to 5 "very high".

Regarding the question "In your opinion, what are the most important objectives defined for the Portugal 2020 Programme?", this is a question with randomised answers where participants could choose up to three options. The options concerned were the following: strengthening research, technological development and innovation; enhancing access to, use and quality of information technologies and communication technologies (ICTs); enhancing the competitiveness of small and medium-sized enterprises-SMEs; promoting sustainable transport and removing bottlenecks in network infrastructures; enhancing institutional capacity of public authorities and the public administration efficiency; promoting sustainability and employment quality; promoting social inclusion, combating poverty and discrimination; investing in education and vocational training for skills acquisition and in lifelong learning; supporting the transition to a low-carbon economy in all sectors; promoting climate change adaptation, risk prevention and management; and preserving and protecting the environment and promoting efficiency in the use of resources.

Finally, regarding the question "what factors may negatively interfere in the future of Portugal and the European Union?", it was an optional question that allowed several answers among the following options: BREXIT; ageing population; extremist ideologies; labour shortages; migratory movements; integration of new countries into the EU; public debt; international economy; unemployment; pandemic of COVID-19 and other infectious diseases; and other (specify).

*3.4. Data Analysis Procedure*

After data collection using the Forms UA platform, they were exported to SPSS Statistics 25 software. Descriptive statistics were used for all questions. For the questions that were considered important to test whether they were independent or not, the chi-square test was used after the respective assumptions were checked.

## 4. Results

The first question asked of the participants was whether they know the Portugal 2020 programme, to which 886 (79.2%) answered affirmatively, 109 (9.7%) answered that they did not know about it and 124 (11.1%) that they were not sure.

Regarding the question "how much do you believe that the programmes financed by European funds contribute to the development of Portugal?", 27 (2.4%) of the participants answered that they believed very little, 60 (5.4%) that they believed a little, 286 (25.6%) that they believed more or less, 524 (46.8%) that they believed sufficiently and 222 (19.8%) that they believed very much.

We then tested whether recognising the Portugal 2020 programme was independent from the fact of believing that programmes financed by European funds contribute to Portugal's development, using the chi-square test. The results indicate that these two variables are not independent ($\gamma^2(8) = 69.93$; $p < 0.001$). The majority of the participants who know the programme believe that the programmes financed by European funds contribute "sufficiently" or "very much" to the development of Portugal (70.1%). However, most of the participants who did not know the program responded with "more or less" or "sufficiently" (69.7%). For those participants who were not sure whether they knew the programme or not, the majority replied that the programme contributed "sufficiently" (51.6%) to the development of Portugal (Table 4).

**Table 4.** How much do you believe that the programmes financed by European funds contribute to Portugal's development? * Do you know Portugal 2020?

| | | Do You Know Portugal 2020? | | | Total |
|---|---|---|---|---|---|
| | | Yes | No | I'm Not Sure | |
| How much do you believe the programmes financed by European funds contribute to Portugal's development? | Very little | 13<br>1.5% | 12<br>11.0% | 2<br>1.6% | 27<br>2.4% |
| | Little | 39<br>4.4% | 12<br>11.0% | 9<br>7.3% | 60<br>5.4% |
| | More or less | 213<br>24.0% | 37<br>33.9% | 36<br>29.0% | 286<br>25.6% |
| | Sufficiently | 421<br>47.5% | 39<br>35.8% | 64<br>51.6% | 524<br>46.8% |
| | Very much | 200<br>22.6% | 9<br>8.3% | 13<br>10.5% | 222<br>19.8% |
| Total | | 886<br>100.0% | 109<br>100.0% | 124<br>100.0% | 1119<br>100.0% |

* Cross-tabulation of variables; Source: Own elaboration.

Consequently, these results validated our research hypothesis H1—the credibility of the contribution of programmes financed by European funds to Portugal's development is not independent of recognition of the Portugal 2020 partnership agreement, with participants who are familiar with this agreement being expected to have higher levels of credibility.

When the participants were asked whether the resources of the Portugal 2020 programme are important for the economic growth of their region and district, 852 (76.1%) answered yes, 55 (4.9%) no and 212 (18.9%) were not sure.

We tested whether the fact of believing that the programmes financed by European funds contribute to the development of Portugal is independent from the fact of considering that the resources of Portugal 2020 are important for the economic growth of their region and their district through the chi-square test, which indicated that these two variables are not independent ($\gamma^2(8) = 379.56$; $p < 0.001$). The participants who consider that the resources of Portugal 2020 are important for the economic growth of their region and their district are also those who mostly believe "sufficiently" or "very much" that the programmes financed by European funds contribute to the development of Portugal (78.3%). As for those who do not consider these resources important, the majority replied "little" or "more or less" (61.8%). Those who are not sure responded mostly "more or less" (52.4%) (Table 5).

Therefore, these results validated our research hypothesis H2—the credibility of the contribution of the programmes financed by European funds to the development of Portugal is not independent of the fact of considering that the resources of Portugal 2020 are important for the economic growth of their region and their district, being expected that the participants who believe more in these programmes are also those who consider that these resources are important for the economic growth of their region and their district.

**Table 5.** How much do you believe that the programmes financed by European funds contribute to the development of Portugal? * Do you consider that the resources of Portugal 2020 are important for the economic growth of your region and your district?

| | | Do you Consider that the Resources of Portugal 2020 are Important for the Economic Growth of your Region and your District? | | | Total |
|---|---|---|---|---|---|
| | | Yes | No | I'm Not Sure | |
| How much do you believe the programmes financed by European funds contribute to Portugal's development? | Very little | 5 0.6% | 14 25.5% | 8 3.8% | 27 2.4% |
| | Little | 22 2.6% | 17 30.9% | 21 9.9% | 60 5.4% |
| | More or less | 158 18.5% | 17 30.9% | 111 52.4% | 286 25.6% |
| | Sufficiently | 456 53.5% | 6 10.9% | 62 29.2% | 524 46.8% |
| | Very much | 211 24.8% | 1 1.8% | 10 4.7% | 222 19.8% |
| Total | | 852 100.0% | 55 100.0% | 212 100.0% | 1119 100.0% |

\* Cross-tabulation of variables; Source: Own elaboration.

Regarding the question about which are the most important objectives of the Portugal 2020 Partnership Agreement, a pulverization of the priority of the chosen objectives is observed since none of them reached even the preference of 50% among the participants. The three most voted objectives were strengthening research, technological development and innovation; enhancing the competitiveness of small and medium enterprises—SMEs (both part of the thematic axis "competitiveness and internationalisation"); and investing in education and vocational training for skills acquisition and in lifelong learning (thematic area "human capital"). In turn, the three less-voted objectives were enhancing institutional capacity of public authorities and Public Administration efficiency; promoting sustainable transport and removing bottlenecks in network infrastructures; and enhancing access to, use and quality of information and communication technologies (ICTs), all three from the thematic axis "competitiveness and internationalisation" (Table 6 and Scheme 1).

**Table 6.** Distribution of the objectives of the Portugal 2020 partnership agreement.

| Objectives of the Portugal 2020 Partnership Agreement | Frequency | Percentage |
|---|---|---|
| TO 1. Strengthening research, technological development and innovation | 453 | 40.5% |
| TO 3. Enhancing the competitiveness of small and medium-sized Enterprises—SMEs | 422 | 37.7% |
| TO 10. Investing in education and vocational training for skills acquisition and in lifelong learning | 404 | 36.1% |
| TO 9. Promoting social inclusion, combating poverty and discrimination | 393 | 35.1% |
| TO 8. Promoting sustainability and employment quality | 370 | 33.1% |
| TO 6. Preserving and protecting the environment and promoting efficiency in the use of resources | 342 | 30.6% |
| TO 5. Promoting climate change adaptation, risk prevention and management | 204 | 18.2% |
| TO 4. Supporting the transition to a low-carbon economy in all sectors | 174 | 15.5% |
| TO 2. Enhancing access to, use and quality of information and communication technologies (ICTs) | 171 | 15.3% |
| TO 7. Promoting sustainable transport and removing bottlenecks in network infrastructures | 149 | 13.3% |
| TO 11. Enhancing institutional capacity of public authorities and the Public Administration efficiency | 147 | 13.1% |

Source: Own elaboration.

The responses on the importance of the 11 thematic objectives were then grouped according to the thematic axes of the Partnership Agreement to which they belong in order to understand the degree of importance attributed to each axis, as shown in Scheme 2. To this end, as the number of objectives varied from axis to axis, the average score of the percentages for each axis was calculated. The axis "human capital" comes first, followed by "social inclusion and employment", "competitiveness and internationalisation" and "sustainability and efficiency in the use of resources".

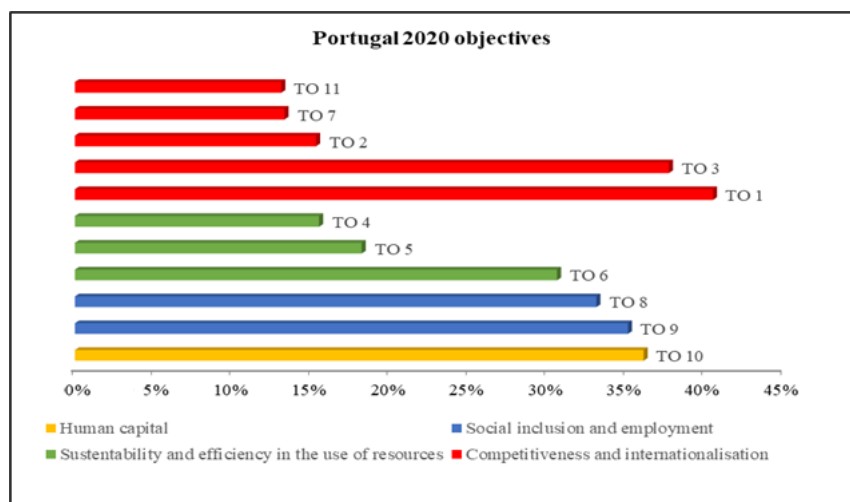

**Scheme 1.** Thematic objectives of Portugal 2020 by axes. Source: Own elaboration.

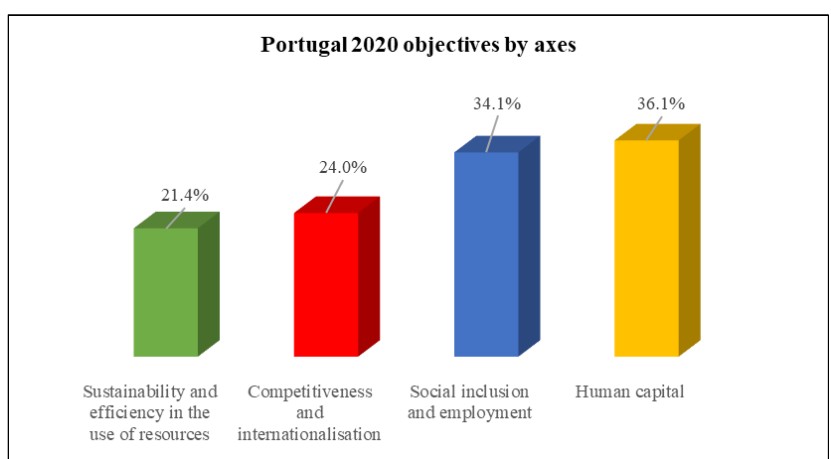

**Scheme 2.** Distribution of Portugal 2020 objectives by axes. Source: Own elaboration.

With respect to the degree of trust in the approval of projects financed by European funds, with regard to adequacy and transparency, 45 (4%) of the participants stated that their degree of trust was very low, 154 (13.8%) that it was low, 577 (51.6%) that it was medium, 306 (27.3%) that it was high and 37 (3.3%) that it was very high. As to whether or not they considered there to be corruption in the allocation of European funds to Portugal, 597 (53.4%) said yes, 132 (11.8%) said no, 373 (33.3%) said they were not sure and 17 (1.5%) made other comments, considering that there may be favouritism, "lobbying", impartiality in the approval of projects, and party preferences, among others.

Similarly, the chi-square test was performed to ascertain whether the degree of confidence in the approval of projects financed by European funds, regarding the adequacy and transparency of the participants is independent or not of the fact of believing that there may be corruption in the allocation of European funds to Portugal. The chi-square test showed that these two variables are not independent ($\gamma^2(12) = 341.70$; $p < 0.001$).

According to Table 7, the data found allow us to state that the participants who believe that there is corruption in the allocation of European funds ($n = 597$) have a mostly medium degree of confidence in the adequacy of the allocation of funds (55.4% of the participants). Of those who do not believe that there is corruption in the allocation of funds ($n = 132$), a significant number have a high or very high degree of confidence in the adequacy of the allocation of funds (80.3% of the 132 participants). In turn, for those who are not sure about corruption in the allocation of funds ($n = 373$), the degree of confidence in the adequate allocation of funds is predominantly medium or high (58.4% and 33.8% of participants,

respectively). Amongst those who responded "other" (*n* = 17), the degree of confidence is mostly between medium (35.3%) and high (52.9%).

**Table 7.** How confident are you in the approval of projects financed by European funds with regard to their adequacy and transparency? * Do you believe that there may be corruption in the allocation of European funds to Portugal?

| | | Do you Believe there May be Corruption in the Allocation of European Funds to Portugal? | | | | Total |
|---|---|---|---|---|---|---|
| | | Yes | No | I'm not sure | Other | |
| How confident are you that projects financed by European funds are approved as to their adequacy and transparency? | Very low | 40 6.7% | 1 0.8% | 3 0.8% | 1 5.9% | 45 4.0% |
| | Bass | 131 21.9% | 3 2.3% | 19 5.1% | 1 5.9% | 154 13.8% |
| | Medium | 331 55.4% | 22 16.7% | 218 58.4% | 6 35.3% | 577 51.6% |
| | High | 91 15.2% | 80 60.6% | 126 33.8% | 9 52.9% | 306 27.3% |
| | Very high | 4 0.7% | 26 19.7% | 7 1.9% | 0 0.0% | 37 3.3% |
| Total | | 597 100.0% | 132 100.0% | 373 100.0% | 17 100.0% | 1119 100.0% |

* Cross-tabulation of variables; Source: Own elaboration.

As a consequence, these results validated our research hypothesis H3—the degree of trust in the approval of projects financed by European funds, in relation to the adequacy and transparency of the participants, is not independent of the fact that in believing that there may be corruption in the allocation of European funds to Portugal, it is expected that the degree of trust in the approval of projects financed by European funds, in relation to the adequacy and transparency of the participants who believe in the existence of corruption, is lower than that of those who do not believe in the existence of corruption.

When asked about the factors that could interfere negatively in the future of Portugal and the European Union, a significant part of the respondents (57.7%) referred to the "COVID-19 pandemic", followed by "public indebtedness" (53.4%) and the "ageing population" (52%). Other concerns that were tangential are "extremist ideologies" (44.1%) and "unemployment" (43.6%). Apparently, the "integration of new countries into the European Union" is not seen with concern (only 3.3% of respondents pointed this factor out), as shown in the Table 8.

**Table 8.** Factors impacting negatively on the future of Portugal and the EU.

| Factors that May Negatively Interfere in the Future of Portugal and the European Union | Frequency | Percentage |
|---|---|---|
| Pandemic COVID-19 and other infectious diseases | 646 | 57.7% |
| Public indebtedness | 598 | 53.4% |
| Population ageing | 582 | 52% |
| Extremist ideologies | 493 | 44.1% |
| Unemployment | 488 | 43.6% |
| BREXIT | 356 | 31.8% |
| International economy | 253 | 22.6% |
| Migratory movements | 138 | 12.3% |
| Labour shortages | 129 | 11.5% |
| Integration of new countries into the EU | 37 | 3.3% |

Source: Own elaboration.

Some respondents (*n* = 44) pointed to other factors that may negatively impact the future of Portugal and the EU (table X), highlighting climate change and corruption, and to a lesser extent, low wage levels, lack of access to formal education and poor preparation of the political class.

## 5. Discussion

Regarding people having knowledge and information about the Portugal 2020 programme, we can infer that the vast majority of citizens who responded to the survey (79%) identified PT2020 and are familiar with its brand. Making a counterpoint with the goal of strengthening the brand and the notoriety of the Portugal 2020 PA [43], our findings provide evidence that this goal may have been achieved since 79% of the participants recognized Portugal 2020.

With regard to the belief in the importance of the EU funds for the development of Portugal, approximately 67% of the participants in this study considered it quite or very important. An even higher proportion (76%) considered that the funds contributed to the development of the region where they live. Our findings for these questions may be considered positive and meet the findings of the AD&C survey [43] that the public recognises the relevance of the European funds for Portugal and regional development. Moreover, the perception that European funds promote the development of EU members is in line with the literature that, under different methodologies, programming periods and territorial coverage, finds positive effects of ESIF to foster the objectives of the EU Cohesion Policy [5,8,10–12,14,16–29,42].

Referring specifically to the Portuguese studies, our findings corroborate the perspective that European funds, since Portugal's entry into the EU, have been an important instrument for development, economic growth and improving the living conditions of the Portuguese population [3,7,37–41].

The assumption that Portuguese citizens recognise the importance of the EU cohesion policy and the benefits that it has brought to Portugal can be evidenced with the results of the last Eurobarometer applied by the EC [45] in which 84% of the Portuguese rejected the idea that the country could better face the future outside the EU.

Regarding the agenda of the Portugal 2020 objectives and axes, the fact that participants listed "strengthening research, technological development and innovation" as the first priority may be associated with the high number of participants from academia, including university professors, researchers and higher education students. The second priority listed was "enhancing the competitiveness of SMEs", belonging to the "competitiveness and internationalisation" axis as the former objective. This finding was to be expected in view of the high priority given to this axis in the absolute values of the funds allocated [6]. What is striking in the results is the little importance attributed to the objectives of "enhancing access, use and quality of ICTs", "promoting sustainable transport" and "strengthening the institutional capacity and the Public Administration efficiency". Perhaps the participants, due to their high level of academic qualifications, have already developed sufficient digital skills and do not consider it as a priority. Still, ICTs will become increasingly necessary and important (the pandemic context in which we live is proof of this—were it not for ICTs, we would not have been able to adapt on a large scale to remote working, distance learning, and online services provision, just to name a few challenges we have had to face in successive lockdowns). Even the Europe 30 Agenda, which will guide the next multiannual financial framework of the EU members, has digitalisation as one of its pillars. In our view, and according to the literature [11,12,20,30], strengthening the institutional capacity of public administrations is essential to ensure a good absorption of funds. In addition, civil servants are expected to be more and more service-skilled, especially with digital skills, to respond efficiently to citizens' needs.

On the other hand, when analysing the average score of the percentages for each axis, in first place appears the axis "human capital", followed by "social inclusion and employment", "competitiveness and internationalisation" and "sustainability and efficiency in the use of resources". Despite the high concentration of Portugal 2020 resources in the axis "competitiveness and internationalisation" [6], participants prioritise human capital and social inclusion and employment axes. These findings are in line with the literature, where there is evidence that investments in human capital and education have longer lasting positive effects [5,20,26,29,42]. Moreover, according to Hofstede [48], Portuguese culture is

very collectivist with marked femininity by valuing human relationships more and having compassion for the less fortunate. This may explain the fact that the study participants prioritise the axis of human capital and social inclusion to the detriment of competitiveness and internationalisation, which may be more related to a masculine culture.

With regard to the degree of the participants' confidence in the adequacy and transparency in the allocation of funds, the percentage who consider it high or very high (30.6%) is greater than those who consider it low or very low (14.2%), with the majority concentrated on a medium degree (51.6%). At the same time, only 11.8% of the participants believe that there is no corruption in the allocation of funds, and 53.4% that there is corruption in it, which is a fairly high portion of the participants' sample. In other comments, participants refer that there may be favouritism, "lobbying", and impartiality in the approval of projects. Could this finding be attributed to a culture of personal relationships historically existing in Portugal? Solomon and Schell [49] attribute to Portugal a score between 18 and 21 (in a score from 0 to 25) of the weight of interpersonal relationships in the business world. The higher the country's score, the more prevalent in its culture are long-term relationships and friendships, the need for trust and the preference for face-to-face meetings to establish business relationships.

The Portuguese even have a special word to describe personal favouring to get a job or otherwise get anything done in a country where everything takes its own time. The word is "Cunha". Everyone knows what it means in Portugal. Namely, "he got his job because he had a Cunha" means that the job was attributed due to a privileged contact that the candidate had. This is commonplace in Portugal—where one possible explanation for Portugal's woes in recent decades has to do with "who you know" being more important than "what you know" —in a culture which does not seem to know the meaning of a "meritocracy" (progression based on merit) in its fullness.

Sousa [50] states that in Portugal, small-scale influence peddling is a systemic practice, the result of favouritism, personal relationships, the art of "desenrascanço" (it is another colloquial Portuguese expression, meaning the ability to solve problems quickly and with few means). As there is a set of practices that are unregulated or difficult to regulate, leaving it more in the scope of ethics to judge these issues, people are inclined to tolerate these practices, albeit recourse to small-scale influence peddling tends to reinforce social inequality [50]. Moreover, in the specific case of European funds, would not the practice of lobbying and influence peddling, in any proportion, be condemning meritocracy and competence, insofar as it may encourage the allocation of resources without legitimacy?

In Javidan [51], a performance orientation (PO) is said to be fundamental in the community in that it shows how the community encourages and rewards innovation (including the perception that there exists a meritocratic environment). The index for the PO dimension is led by Switzerland, with Singapore following (practices in society). Portugal appears in level C, the lowest, at eight places from the bottom [51]. Hence, our study confirms what the literature states may be occurring—namely that in Portugal, a performance orientation, or a meritocracy, is lacking.

Therefore, drawing a parallel to the attribution of millions of euros in subsidies, it may be more than natural that the Portuguese see in the subsidies an opportunity for those in power to favour their friends and associates rather than attributing funds according to those who merit or need them the most, hence the interest of this study—to determine the perceptions of the Portugal 2020 programme by Portuguese citizens. Do they perceive foul play in the attribution of funds?

The majority of the respondents to the survey believe in the Portugal 2020 programme. However, and as perhaps expected, the perception of corruption (or simply the "Cunha" culture) is high.

As stated in the results section, the three research hypotheses were confirmed with our findings, which can validate our proposed conceptual model. The respondents who recognize and identify Portugal 2020 as an EU programme (and brand) have a more positive perception of the benefits of European funds for Portugal (70.1%). Furthermore,

respondents who have a positive perception of the benefits of the funds for the country also have a greater appreciation that the funds contribute to the economic growth of the regions in Portugal (78.3%). Additionally, the majority of the respondents who believe that there may be corruption in the ESIF allocation (53.4%) have a low or medium appreciation of the adequacy and transparency in the approval of funded projects (73.3%).

Regarding the factors that may negatively impact the future of Portugal and the European Union, as predicted, the pandemic of COVID-19 appears in first position. Pointed out as the second factor, Portugal's public indebtedness will grow as the severe global economic crisis coming with the pandemic will further hit countries with structurally weaker economies, corroborating Malakhova et al. [13]. These findings are in line with the last EC Eurobarometer [45], where the Portuguese showed great concern about the collateral effects of the pandemic coronavirus on Portugal's economy and their financial situation, besides being totally hopeless in a short-term economic recovery. Population ageing appears as a third negative factor in our study. Population ageing is such a growing concern in the European agenda, and common to a lesser or greater extent to all EU members, that the new Europe 2030 strategy emphasises demographic recomposition as one of the crucial objectives.

## 6. Conclusions

We consider that our research questions were answered and shed some light on the relevant debate on the perception of the effects of structural funds on EU economies. Portugal, in particular, presents a more vulnerable position in the European context, and due to having regions falling under Objective 1 of convergence, receives large amounts of structural funds. Despite the results pointing to a positive perception of the impact of the funds on the development of the country and its regions, it is necessary to question whether the resources are well used and whether there is adequacy and transparency in the allocation of funds. In this sense, the greatest contribution of this study, in the authors' views, was learning that more than half the participants believe that there may be corruption in the allocation of funds. Despite the cultural factors discussed in the previous section, this finding should be analysed carefully and deserves further investigation.

In addition, the study provided insight into society's concerns about the problems that could endanger the European project and the future of the EU members. If the questionnaire had been applied before the pandemic of COVID-19, the results concerning threats to the stability of the EU might have been different. Possibly, we would not even have foreseen the possibility of an infectious disease pandemic as such a risk factor.

Nevertheless, despite presenting meaningful contributions to the research issues, the study has some limitations. One of them concerns the high number of participants with high academic qualifications, and some degree of bias is expected in this context. The study was widespread in universities, polytechnics, schools, companies and trade associations, parish councils of all districts of Portugal, beyond a few social media channels. Apparently, the academic environment, due to its end activity of teaching and research, is more inclined to contribute with research projects and participate in surveys.

Another limitation encountered by the authors after data processing was the impossibility in associating sociodemographic data of the participants by region at the NUTS 2 and NUTS 3 level since the data adopted in the study were the district of residence and a district may fall within more than one NUTS classification. For example, the district of Aveiro has councils classified both in the Central Region and in the Northern Region (at the NUTS 2 level) and in the Aveiro Region, Metropolitan Area of Porto, Tâmega and Sousa (at the NUTS 3 level). We recommend for those who need a territorial approach at the NUTS 2 or 3 level to inquire about the council of residence instead of the district.

In this study, correlational analyses of the variables with sociodemographic data were not applied to check how survey responses behave according to groups. Future studies may apply this statistical approach, and others. Another possible line of research is to

compare the findings of this study with the results of the opinion survey that will be carried out by the Portugal 2020 Management Authority (AD&C).

Finally, we hope that our findings circumspect in this study may contribute in some way to the definition of the policies to be implemented for the next ESIF programming period of 2021–2027. The Portugal 2030 Partnership Agreement is currently being drafted and will be based, at the thematic level, on three areas: demography and inclusion; innovation and digital transition; and climate transition and sustainability of resources.

**Author Contributions:** Conceptualization, A.Z.F.C.N., A.M., M.A.-Y.-O. and M.J.S.; methodology, A.Z.F.C.N., A.M., M.A.-Y.-O. and M.J.S.; investigation: A.Z.F.C.N., M.A.-Y.-O.; software: A.Z.F.C.N., A.M., M.A.-Y.-O. and M.J.S.; validation, A.M., M.A.-Y.-O. and M.J.S.; writing—original draft preparation, A.Z.F.C.N., M.A.-Y.-O.; writing—review and editing, A.Z.F.C.N., A.M., M.A.-Y.-O. and M.J.S.; supervision, A.M., M.A.-Y.-O. and M.J.S.; funding, M.A.-Y.-O. All authors have read and agreed to the published version of the manuscript.

**Funding:** This research received no external funding.

**Institutional Review Board Statement:** Not applicable.

**Informed Consent Statement:** Not applicable.

**Data Availability Statement:** Not applicable.

**Acknowledgments:** The authors would like to thank the valuable comments received from three anonymous reviewers from the reviewer board of the journal. The authors also acknowledge the contribution of the participants who answered the survey.

**Conflicts of Interest:** The authors declare no conflict of interest.

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
