# Peer review of "Effectiveness of the Portugal 2020 Programme: A Study from the Citizens’ Perspective"

_sustainability, doi:10.3390/su13115799_

Round 1
Reviewer 1 Report
Dear Authors,
I am impressed by the number of respondents who took part in the survey you conducted. I also find the topic of the study relevant - issues related to the cohesion policy seem to be a theme that is difficult to exhaust. However, I have some concerns about the reviewed work, which I will try to present below. I will begin with some more general comments, followed by more specific observations on issues of perhaps lesser importance.
My primary comment concerns the question of what is the paper really about? What is the main research problem and what is its real purpose? Is it to be believed that, as the title suggests, it is about smart, sustainable and inclusive growth? If so, this objective is not reflected in the content. The description of the objectives given in lines 120-126 is appropriate, but information about it should have been outlined already in the abstract. Thus, while I acknowledge that the stated aims are adequate to the content and the empirical layer, I consider the lack of logical connection between the aims of the paper (research questions) and the hypotheses to be a significant weakness. They are formulated as if "next to" the main objectives, in fact, it is not clear what for. This is evidenced by the fact that the issue of verification of research hypotheses is not addressed in the "discussion" section. Their verification is only outlined in the "results" section.
Secondly, although the structure of the article seems appropriate at first glance, in reality, the key items (introduction, literature review, results and discussion) seem to be poorly connected - both logically and substantively.
First of all, I do not see the connection between the literature review and the study itself. The literature review (section 2) should provide a certain foundation used to conduct the study and serve to place the obtained results in a certain context. While the study concerns the PERCEPTION of the Portugal 2020 agenda (i.e. something subjective), the literature review presents a broad cross-section of studies based on econometric methods concerning the impact of cohesion policy on economic or employment growth, i.e. issues of an objective nature. Therefore, it would be better to analyse the studies in the context of the objective of the paper - how is cohesion policy perceived and evaluated? What advantages and disadvantages are crucial from the perspective of society? If you intended to juxtapose objective effects vs public perception, this should have been highlighted in the text. However, I believe that the literature review does not really address the research problem undertaken in the study - the perception of the effects of cohesion policy in Portugal.
It is also important to better link the introduction with the subject of the study. The introduction should introduce the reader to the presentation of the study's results. It should encourage the reader to read it. It should point out the essence of the research problem. In this case, the introduction is a not very interesting description of the Portugal 2020 instrument together with a description of its objectives and allocation values. I am not saying that there is no place in the article for a presentation of the Portugal 2020 programme. It is quite important in the context of the study. However, such a description should be included in the text after the literature review. The introduction should provide a real “introduction” to the research problem undertaken in the study.
Specific comments:
- The title needs to be changed. It is not attractive, but above all, it is not in line with the content of the article. The article devotes little space to the issue of smart, sustainable and inclusive growth.
- The abstract should indicate what the aim of the study is. Currently, this is not clear.
- abstract, verse 15 - there is a typo Eu instead of EU
- v. 95-96 - clarification of NUTS - move to footnote
- v. 99 - typo - km2 (upper letter)
- v. 120-142 - it is necessary to link the objectives of the paper to the hypotheses (see above)
- 2 Impacts of ESIF... - this section should be better linked to the objectives of the work. I admit that the table with the results of empirical studies on the impacts of ESIF is valuable and presents a very interesting summary/overview, but what relation does it have to the objectives of the work? What does it lead to? Try to prove it, or give it up. Leaving that aside I also have some reservations about the narrative in the research review section. Each paragraph is effectively a summary of a single study. This is not very interesting. Try to make this review more cross-sectional, try to combine common findings.
- The study is unrepresentative. A big burden is the above-average participation of scientists and researchers. I know this can't be changed now, but it matters in terms of interpreting the results (see further below).
- v. 314-338 - it would be good to refer to the survey questionnaire included as an appendix.
- graph 1 - add a description or numbering of the Portugal 2020 objectives and link the graphic to table 6
- discussion - you start with a description of the diagnosis carried out by AD&C. I understand that you want to use this as a reference point for the results of your own study. However, I suggest that you do not start with this as it is quite confusing. Better to start by recalling the objectives of the study, contrast the results with those of other studies. You can refer to the AD&C study at the end, but its description needs to be more concise. Also, you cannot compare the results of the AD&C study and the results of your study, which are biased by the overrepresentation of people better informed about the objectives of cohesion policy than the general public.
- v. 568-581 - the question of hypothesis verification is devoid of any interpretation. Why then were the hypotheses formulated?
- v. 584 - not effects, but the perception of the effects
- v. 606-611- this is not a conclusion
Author Response
"Please see the attachment."

Reviewer 2 Report
Based on the chosen interview method, the authors investigate the impact of the ESIF on national and regional development effects in Portugal. The results of the research indicate a positive evaluation of the EU funds for the development of Portugal and their regions, although a reasonable percentage of respondents consider that there could be corruption or practices such as "lobbying" in the allocation of the funds. The results obtained and the three initial hypotheses are argued in the discussion. The authors also point out the methodological limitations of the conducted research, as well as possible further areas of research on this topic.
The summary of the paper could be more detailed in terms of the findings obtained. Also, the literature review could be extended in the part related to the study of the impact of ESIF funds on the growth and development of EU countries. However, this is not necessary for the publication of the papar but only for its possible improvement.
Author Response
"Please see the attachment."

Reviewer 3 Report
General Comments
The article debates a relevant issue for the academic debate which relates to the effects of EU cohesion policies in EU member states in several territorial development dimensions. Based on the Portuguese case and using data from enquiries to citizens, the authors identify a number of interesting and relevant conclusions which can help to better revise the effectiveness of the implementation of these policies in future programming periods.
In all, the article is well organised and written. It uses relevant and updated references. The introduction presents the research problem in an appropriate manner. A methodological explanation is presented in after the literature review on the impacts of EU cohesion policy. This section is quite complete. However, some key studies on the impacts of EU Cohesion Policy in Portugal, for instance, are missing (see below proposed references).
The number of participants in the enquire looks appropriate and the explanation on their ages and qualifications is presented. It would be interesting to know the location (NUTS 3) of the participants.
The results section is a bit descriptive, but the results are comprehensively presented in tables.
The discussion section is well-elaborated, and the conclusions are appropriate in view of the obtained data and respective analysis.
In all, I found the article very interesting and comprehensive. It provides useful data to better understand the effeteness of EU Cohesion Policy in Portugal (Portugal 2020 policy framework) from the citizen’s perspectives, which is largely absents from current literature.
Specific Comments
Figure 2: the numbers inside the figure are not particularly visible and clear
All tables need the placement of the source below.
Proposed references:
Crescenzi, R. & Giua, M. (2020) One or many Cohesion Policies of the European Union? On the differential economic impacts of Cohesion Policy across member states, Regional Studies, 54(1): 10-20.
- Medeiros, E. (2014) Assessing territorial impacts of the EU Cohesion Policy: the Portuguese case, European Planning Studies, 22 (9): 1960-1988
Author Response
"Please see the attachment."

Round 2
Reviewer 1 Report
Dear Authors,
I am very satisfied with the changes and amendments you made to the text. In my opinion, the article is suitable for publication as a valuable contribution to cohesion policy research. I also wish you good luck with your PhD thesis.
Best wishes